# Discrimination of Mangrove Stages Using Multitemporal Sentinel-1 C-Band Backscatter and Sentinel-2 Data—A Case Study in Samut Songkhram Province, Thailand

Kamonporn Upakankaew [1,*], Sarawut Ninsawat [1], Salvatore G. P. Virdis [1] and Nophea Sasaki [2]

1   Remote Sensing and Geographic Information Systems, School of Engineering and Technology, Asian Institute of Technology, P.O. Box 4, Klong Luang 12120, Thailand
2   Natural Resources Management, School of Environment, Resources, and Management, Asian Institute of Technology, P.O. Box 4, Klong Luang 12120, Thailand
*   Correspondence: nest06th@gmail.com; Tel.: +66-81-273-5323

**Abstract:** Discrimination of mangrove stage changes is useful for the conservation of this valuable natural resource. However, present-day optical satellite imagery is not fully reliable due to its high sensitivity to weather conditions and tidal variables. Here, we used the Vertical Transmit—Vertical Receive Polarization (VV) and Vertical Transmit—Horizontal Receive Polarization (VH) backscatter from the same and multiple-incidence angles from Sentinel-1 SAR C-band along with Normalized Difference Vegetation Index (NDVI), Normalized Difference Water Index (NDWI), Normalized Difference Moisture Index (NDMI), Normalized Difference Red Edge (NDVI$_{RE}$) and Chlorophyll Index Green (CI$_{Green}$) from the optical satellite imageries from Sentinel-2 to discriminate between the changes in disturbance, recovery, and healthy mangrove stages in Samut Songkhram province, Thailand. We found the mean NDVI values to be 0.08 ($\pm$0.11), 0.19 ($\pm$0.09), and $-$0.53 ($\pm$0.16) for the three stages, respectively. We further found their correlation with VH backscatter from the multiple-incidence angles at about $-$17.98 ($\pm$2.34), $-$16.43 ($\pm$1.59), and $-$13.40 ($\pm$1.07), respectively. The VH backscatter from multiple-incidence angles was correlated with NDVI using Pearson's correlation ($r^2 = 0.62$). However, Pearson's correlation of a single plot (ID2) of mangrove stage change from disturbance to recovery, and then on to the healthy mangrove stage, displayed a $r^2$ of 0.93 (*p* value is less than 0.0001, *n* = 34). This indicated that the multitemporal Sentinel-1 C-band backscatter and Sentinel-2 data could be used to discriminate mangrove stages, and that a reduced correlation to significant observations was the result of variations in both optical and SAR backscatter data.

**Keywords:** multitemporal data; Sentinel-1; C-band; Sentinel-2; mangrove stage discrimination

## 1. Introduction

Ecosystem services of mangrove forests are important for the inhabitants living along these coastal areas, and to the wider population in general. These major ecosystem services include wood for construction and fuel, habitats for coastal fauna, nurseries for juvenile marine organisms, food for humans and animals, income sources from fishery, carbon (C) storage, protection from strong winds during typhoons, and coastal erosion mitigation [1,2]. As referenced in [3], the National Economic and Social Development Plans (NESDP) of Thailand 2011–2013 [4] include several mangrove restorations and conservation projects; indeed, there exist ongoing mangrove restorations in Thailand at the present time. Nevertheless, the loss and diminution of these valuable natural resources still occurs apace [5]. In [2,6,7] the authors discuss that as the mangrove forests continue to decline and degrade, regular assessment needs to be undertaken so that appropriate measures can be introduced to identify their conditional stage. Such monitoring would be the basis for better informed decision-making in pursuit of sustainable conservation, resource management, and the restoration of existing mangroves.

To monitor a mangrove's stage changes (e.g., disturbances and recovery stages), optical remote sensing techniques, such as NDVI and NDMI, are used. Additionally, detailed spatial resolution satellites such as SPOT are employed. These satellites are globally positioned and are highly accurate [8–11]. The studies by Cho and Lee recommend NDVI and NDMI techniques. These are derived from Sentinel-2 data and have a high capacity for discriminating mangroves in particular [8,9,11]. NDVI was widely applied to detect a mangrove forest's healthy conditional changes, while NDMI showed high performance in detecting mangrove disturbance changes [5,10,12]. Furthermore, $NDVI_{RE}$ and $CI_{Green}$ were used to study leaf chlorophyll content, biomass, and map the extent of the mangrove [13,14]. $NDVI_{RE}$ was the vegetation index composited from the red-edge band. The red-edge bands were sensitive to vegetation's biophysical variables (e.g., leaf chlorophyll and water content) [11,14,15]. A study [16] revealed that in extracting the mangrove extent and species from Sentinel-2 imagery, the red-edge bands were the most informative variables in discriminating mangrove species, followed by shortwave infrared and near-infrared bands. Red Edge B5, and its derived $CI_{Green}$ and $NDVI_{RE}$ indices, are ranked first, third, and fourth, respectively. The shortwave infrared B11 was the second most important variable.

Several studies found that robustness of hyperspectral data from optical sensors have a very promising potential in classifying mangrove forest, their species distributions, and even their health conditions [17–20]. Unfortunately, the availability of hyperspectral data from optical imagery is limited because it is highly sensitive to weather conditions, and tidal variables [21,22]. Younes Cardenas et al. (2017) [23] conducted 55 reviewing articles on monitoring the mangrove ecosystem from Landsat/Aster imagery. The authors found that most of the studies between 2001–2016 did not make full use of the wealth of optical data available to analyze the temporal changes of mangrove due to cloud cover; thus, making it unfeasible to monitor mangrove forest changes over the course of an entire year when climatic conditions intervene [24].

The phenological behaviors of mangrove forests can also affect detection using the remote sensing technology. Their phenological behavioral changes can be detected through the time-series of NDVI, NDMI from MODIS sensor, and Landsat satellite [25,26]. The high-spatial resolution satellite imagery is known as best practice for detecting mangrove forest conditions; however, this imaging method is cost prohibitive [27,28]. Consequently, the budget concerns over the costs of high-resolution satellite imagery could understandably lead to insufficient information, which can result in ineffective monitoring of mangrove forests [18].

On the other hand, SAR remote sensing can penetrate almost any weather condition; therefore, can provide more temporal frequency during the day and night [20–31]. The study by Thomas [32] confirmed that the HH polarization channel of JERS-1SAR and ALOS using L-band SAR (PALSAR) was able to monitor the global mangrove deforestation and degradation from 1996 to 2010 by focusing on the drivers of such loss. Due to their wavelength capacities, L-band can more deeply penetrate the dense foliage of mangrove, resulting in a higher accuracy using this technology [32]. However, a single scene of the L band is very costly [33–35], making this method impossible for the broad use for mangrove monitoring, especially in conditions requiring a large volume of high-res temporal images.

With recent technology advancement and data availability, Sentinel-1 [27,33,34] has provided SAR C band images free of charge since 2014 but the images covering Thailand have only been available since late 2015 [30,36,37]. Eventually, the Sentinel-1 SAR C-band has been employed to assess the vegetation behaviors and related change. For example, a study by Nasirzadehdizaji [38] evidenced that Sentinel-1 SAR using VV + VH and VV parameters in the growing season had a high capacity to estimate maize height, especially during the early growing stage. Another study [39] used multitemporal Sentinel-1 to monitor phenology and classify deciduous and coniferous forests in Northern Switzerland. It found that in VH, deciduous species showed a higher backscatter in the winter than in the summer, whereas spruce showed a higher backscatter in the summer than in the

winter. In VV, this pattern was similar for spruce, while no distinct seasonal behavior was apparent for the deciduous species. Although their successful studies provide the fundamental understanding of the roles of remote sensing technology, comparatively fewer studies have attempted to use multitemporal SAR backscatter method to discriminate the mangrove stages.

This study was designed to investigate the potential use of multitemporal and multi-incidence angle Sentinel-1 SAR backscatter on VV and VH polarizations, alongside vegetation indices (such as NDVI, NDMI, $CI_{Green}$, and $NDVI_{RE}$ from Sentinel-2 data), for discriminating mangrove forest stages and their changes, using mangrove forests in Samut Songkhram province, Thailand as a case study. The obtained potential uses of the above remote-sensing technologies are important for monitoring the changes of mangrove forests at different stages, where such changes are affected by natural or anthropogenic activities. The information on change stages is also important for better informed decision-making in the restoration and conservation of the mangrove forests.

## 2. Materials and Methods

### 2.1. Study Area

The mangrove forest selected for this study is located in Samut Songkhram province, which is in the upper Gulf of Thailand (Figure 1). During various time periods in this mangrove, the diverse magnitude of natural and anthropogenic stresses on mangroves led to mangrove disturbance and recovery in different scales and periods in this area [40,41]. Mangrove forests here undergo changes in various stages due to several coastal project developments and urban expansion.

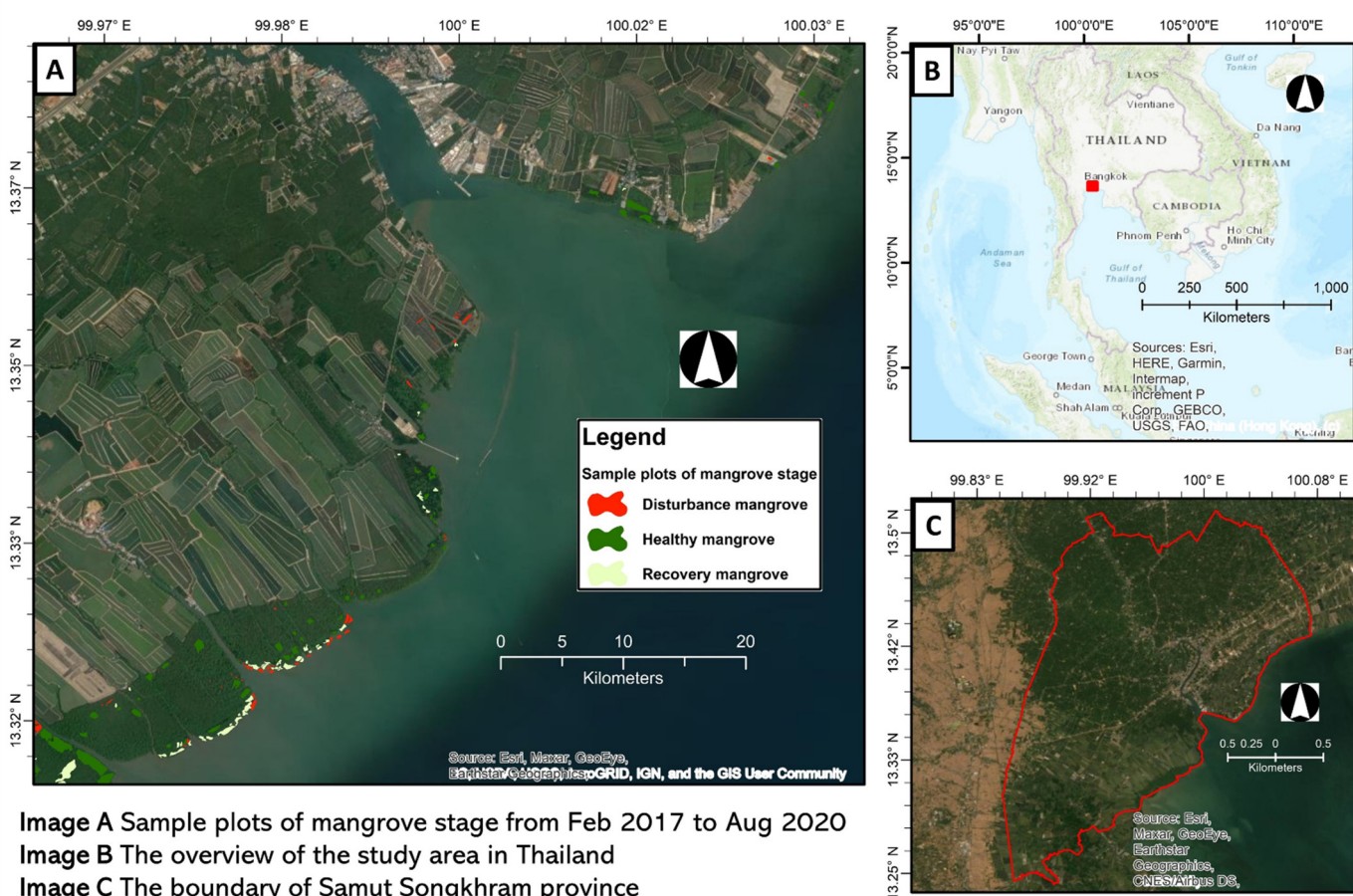

Image A Sample plots of mangrove stage from Feb 2017 to Aug 2020
Image B The overview of the study area in Thailand
Image C The boundary of Samut Songkhram province

**Figure 1.** Study Area.

## 2.2. Satellite Data Preprocessing and Analysis

### 2.2.1. Defining Polygon Samples of Different Mangrove Stages on Google Earth Pro

The available past and current ground-truth data on general mangrove conditions are very rare due to the inconvenience of accessing mangrove forests, the personnel skill set required, and the labor intensity [42]. Richards and Friesse [34] employed multitemporal images from Google Earth Pro to view and identify the mangrove deforestation and regrowth patches, and inferred various changes in land circumstance including aquaculture, rice field, oil palm plantation, urban sprawl, terrestrial forest, and coastal erosion. They found that Google Earth Pro provided similar accuracy for mapping mangrove communities using visual interpretation compared to commercial images such as Rapid Eye satellite (RE) images. The accuracy was even greater when combining Google Earth Pro with NDVI [43]. In congruence with their findings, this study also based the interpretations on Google Earth imagery to identify stages in the sample areas of our subject mangrove.

Our study considered a time-series of mangrove change to identify their stage change. Mangrove stages were classed into three main groups, i.e., disturbance, recovery, and healthy mangrove stages. The size of each polygon was larger than 100 square meters to maximize the spatial resolution of Sentinel-1 imagery. Our interpretations were based primarily on the tonality and texture of mangroves. Other attributes, such as shape and closeness to other features, in conjunction with the tonality and texture attributes, helped to further interpret the images [44,45]. The total number of mangrove sample polygons was 245 overall. Figure 2 illustrates how the different mangrove stages are defined on Google Earth Pro using their historical imageries.

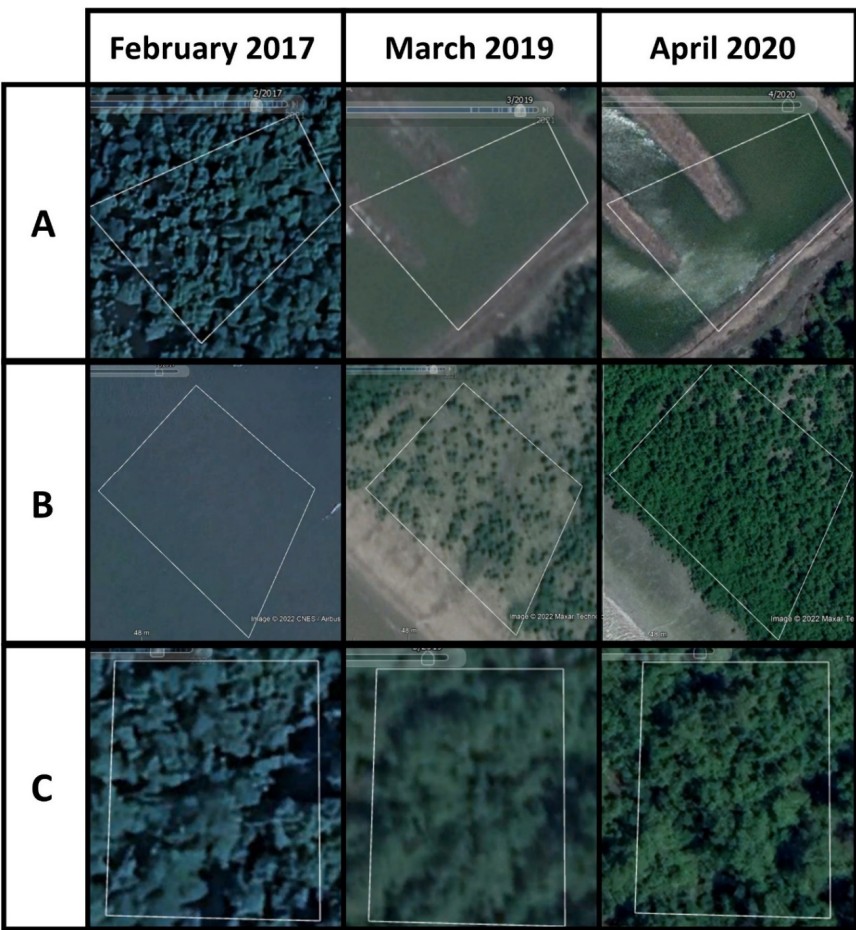

**Figure 2.** Defining Polygon Samples of Different Mangrove Stages on Google Earth Pro Using Their Historical Imageries (**A**) Disturbance Mangrove Stage Polygon Sample (**B**) Recovery Mangrove Stage Polygon Sample (**C**) Healthy Mangrove Stage Polygon Sample.

### 2.2.2. Sentinel-2 Data

Data from the Sentinel-2 Multi-spectral Instrument (MSI) level-1C (Level 1C Top-of Atmosphere reflectance products) were acquired for the period covering January 2016 to August 2020. The specification of the Sentinel-2 Data is illustrated in Supplementary Table S1: The Specification of Sentinel-2 Data. Each vegetation index including NDVI, NDMI, $NDVI_{RE}$ and $CI_{Green}$ derived from Sentinel-2 was processed on the Google Earth Engine (GEE) platform and the results were encapsulated in GeoJSON file format. The overall Sentinel-2 images numbered 495 scenes. The pre-processing started with importing polygons to the GEE platform. Next, we selected satellite images of Sentinel-2 and imported polygons of identified mangrove stages from the GEE platform. The cloud covering in Sentinel-2 scenes was masked using bands 10 and 11. Red edge and SWIR bands, which had a spatial resolution of 20 m, were resampled to 10 m (to match the same spatial resolution of Sentinel-1 SAR) before band composites were made. Then Green, Red, Red edge1, NIR, and SWIR bands were selected for a vegetation index composite using NDVI, NDMI, $NDVI_{RE}$ and $CI_{Green}$. The formula of the vegetation indices used in this study is shown in Table 1.

**Table 1.** Vegetation Indices Formula.

| Name of Vegetation Indices | Formula |
| --- | --- |
| • Normalized Difference Red (NIR)/Normalized Difference Vegetation Index (NDVI) | $(NIR - Red)/(NIR + Red)$ or $(B8 - B4)/(B8 + B4)$ |
| • Normalized Difference Moisture Index (NDMI) | $(SWIR - NIR)/(SWIR + NIR)$ or $(B11 - B8)/(B11 + B8)$ |
| • Normalized Difference Red edge/Red $NDVI_{RE}$) | $(NIR - Red\ edge)/(NIR + Red\ edge)$ or $(B8 - B5)/(B8 + B5)$ |
| • Chlorophyll Index Green ($CI_{Green}$) | $(NIR/Green) - 1$ or $(B8/B3) - 1$ |

Next, zonal statistics were used to extract the averaged 10 maximum random values of vegetation indices. The averaged 10 maximum random values were used because they can provide higher values than others, such as the averaged vegetation indices of each polygon. This might be the result of some values which were mixing with soil or moisture reflectance. In addition, we assigned 10 random points because we proposed the minimum size of polygon was 100 square meters for this study.

Then, these vegetation indices were grouped monthly, thus, some of the missing pixels were filled in this step. The data were exported using GeoJson format for converting to CSV format in the Jupyter notebook. Some of the missing data were not filled in this study to observe the amount of missing data from Sentinel-2 and invest their correlation to order to fill the gap of missing values from atmospheric variability from optical sensors [46].

### 2.2.3. Sentinel-1 Data

Accessing via the Copernicus Sentinel Scientific Data Hub, this study used the Sentinel-1 C-band level-1 Ground Range Detected High Resolution (GRDH) product from both an ascending and descending pass with a 10 m spatial resolution. We investigated both the Vertical transmit—Horizontal (VH) and Vertical transmit—Vertical (VV) polarizations. The relative orbit numbers covering the study area were 62 and 172. Both orbits had nominal incident range between 32.9–38.3°, or, alternatively, Interferometric Wide swath 1 (IW1) and Interferometric Wide swath 2 (IW2), respectively. As the 2015 study [47] indicated, the incidence angle of the C-band image was effective in detecting parts of trees, with the larger incidence angle being better suited at detecting the vertical tree components (i.e., tree trunks and branches) as they presented a greater surface area to the radar. Moreover, a smaller incidence angle tended to have a greater interaction with the upper canopy and hence, greater attenuation by the branches. Due to fears over signal variations causing data misinterpretation, it was found advisable that the data be analyzed using the same sensor characteristics. [48]. As such, to reduce the effects of C-band incidence angle and

their varying backscatter, this study considered the influence of incidence angle together with pass direction to investigate their efficacy on the discrimination of mangrove stages.

Sentinel-1 SAR data covering the study areas are available from both ascending and descending direction modes (Supplementary Figure S1: Sentinel-1 C-band Fly Directions Over the Study Area). However, the SAR data from ascending mode were only made available as of February 2017, while the data from descending mode were available since October 2014. To comparatively analyze variables between Sentinel-1 SAR and Sentinel-2, the correlation analysis between the two data sources was started in February 2017. There were approximately 257 images made available at that time, with 105 images in ascending mode and 152 in descending mode. Image processing consists of five steps and is designed for optimal reduction of error propagation in the subsequent processes [49]; to obtain precise orbits, we downloaded the orbit file and updated the orbit state vectors from the product's metadata. A subset was then recommended to reduce the processing time. Next, we converted digital pixel values to radiometrically calibrated SAR backscatter. Specifically, this is a calibration vector included as an annotation in the product and allowed the simple conversion of image intensity values into sigma nought values. This was used because most coastal areas around the Gulf of Thailand do not vary much in topography and Range Doppler Terrain Correction is sufficient to geolocate the data to a common spatial grid. The last step of image preprocessing was to convert the backscatter coefficient to dB using a logarithmic transformation. Finally, six SAR variables are illustrated below (Figure 3).

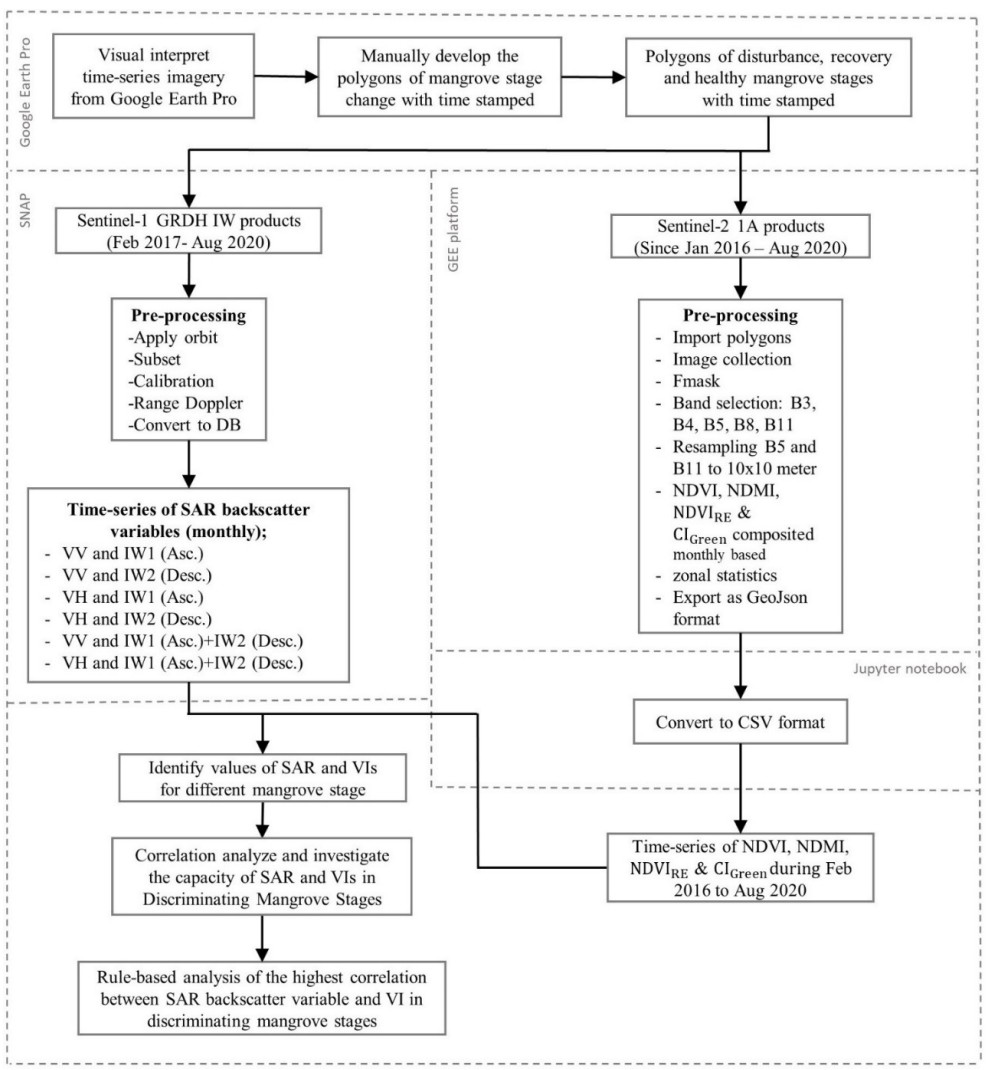

**Figure 3.** Overall Study Workflow.

2.2.4. Discrimination of Mangrove Stages using Vegetation Indices and SAR Backscatter Variables

Discrimination of Mangrove Stages from Previous Studies

To analyze the correlation between the vegetation indices and SAR backscatter, the values of different vegetation indices and SAR backscatter of each mangrove stage needed to be defined. The values of the vegetation indices and SAR Backscatter from previous studies are illustrated in Tables 2 and 3, respectively.

**Table 2.** Values of Vegetation Indices for Discrimination of Mangrove Stages/Previous Studies.

| Mangrove Stages | NDVI | References | NDMI | References | $NDVI_{RE}$ | References | $CI_{Green}$ | References |
|---|---|---|---|---|---|---|---|---|
| Disturbance | ≤0.30 | [50–52] | - | - | - | - | - | - |
| Healthy | >0.70 | [53,54] | 0.10–0.80 | [8,9] | 0.07–0.16 | [55] | 1.50–7.00 | [56] |
| Recovery | 0.10–0.30 | [50–52] | - | - | - | - | - | - |

**Table 3.** The Values of VH and VV backscatter for Discrimination of Mangrove Stages/Previous Studies.

| Mangrove Stages | VH | References | VV | References |
|---|---|---|---|---|
| Disturbance | −16 to −26 | [47] | - | - |
| Healthy | −14 to −10 | [47] | −8 to −10 | [57] |
| | −14 to −15 | [57] | - | - |
| Recovery | −16 to −26 | [47] | - | - |

Rule-Based Discrimination of Mangrove Stages

According to SAR, the C-band cannot penetrate a forest canopy that is too complex. It results in early saturation when compared with NDVI [58,59]. As such, we proposed rule-basing to discriminate the mangrove stages. Rule-basing considers the capacity of the vegetation indices and SAR backscatter on time-series. The values of NDVI, NDMI, $NDVI_{RE}$ and $CI_{Green}$ higher than 0.3, 0.1, 0.07 and 1.5, respectively, were regarded as healthy mangroves.

NDMI, $NDVI_{RE}$ and $CI_{Green}$ were rarely studied for disturbance or recovery mangrove stages. To address this lack of data for the recovery mangrove stage, NDVI from previous studies were used [50–52] combined with past observations of mangrove stages. We can thus extrapolate that if the NDVI in the past was lower than the NDVI in recent times, a conclusion can be drawn that the mangrove is in its recovery stage.

In addition, disturbance is a mangrove condition that provides a lower capacity to supply products and services [60]. In remote sensing, forest disturbance can be monitored using indicators such as time-series and biomass related to vegetation indices from optical sensors [50–52]. It should be noted that the subject mangrove here in Thailand is sparse and has a NDVI around 0.1–0.3. The NDMI value for a mangrove forest in disturbance was less than 0.1 [8,9]. Due to fewer studies completed on the values of the mangroves in disturbance stages using $NDVI_{RE}$ and $CI_{Green}$, their disturbance values were simply defined as the values that were less than their healthy mangrove stages. Therefore, in this study, the samples of NDVI, NDMI, $NDVI_{RE}$ and $CI_{Green}$ that were less than 0.3, 0.1, 0.07, and 1.5, respectively, and based on their time-series history or an earlier higher value of mangrove stage, were to be regarded as a disturbance mangrove.

We also considered SAR backscatter and time-series to identify the disturbance and recovery mangrove stages. A 2015 study [47] used Radarsat-2 (C-band) to model the biophysical parameters of regenerating mangroves. The incidence angle of Radarsat-2 was 23.39–25.28 degrees. They found that the values of VH were about −26 to −16 dB with the mean of all plots being −17 dB. These values were used as a reference for our recovery mangrove stage. Additionally, the SAR backscatter before the present recovery mangrove stage must be lower than this value. The SAR backscatter reference values of the

disturbance mangroves were based on the mean values of VH, which were about $-26$ to $-16$ dB (similar to the recovery mangrove stage), with the difference being that the SAR backscatter values must have been higher before the mangroves were disturbed.

### 2.2.5. Analysis of Rule-Based Correlation and Capacity

Correlation of Mangrove Stage Discrimination Analysis

Correlation analysis was conducted on the data collected for the period February 2017 to August 2020. Monthly vegetation indices and SAR backscatter variables were analyzed to find their correlation. Different models were applied to find the appropriate analytical tool, e.g., linear regression, polynomial regression, and Pearson correlation coefficient.

Capacity of Rule-Based Mangrove Stage Discrimination Analysis

We analyzed the capacity of each parameter (from vegetation indices and SAR backscatter) for mangrove stage discrimination using rule-based methodology and observed their matching on time-series. To analyze how the rule-based works methodology when discriminating the mangrove stages on time-series data, we labeled the values of each parameter on the time series to identify the mangrove stage changes at their specific time of change. Here, we also observed the change in values of vegetation indices and SAR backscatter variables in the different mangrove stages to assess their capacity in discriminating mangrove stages.

### 2.2.6. Most Frequently Correlated Variables in Discriminating Mangrove Stages Analysis

We analyzed a pair of the most frequently correlated parameters with respect to their capacity for performance and limitations in the discrimination of mangrove stages. Both the vegetation index and SAR variables were investigated in plots which were representative of different mangrove stages; then, we labeled them using rule-based methodology on time-series data. Here, we could observe which variable had a higher capacity, and which variable was more limited in discriminating mangrove stages. Furthermore, the rule-based methodology for different mangrove stage discrimination was analyzed to assess the capacity on time-series data using both clusters and density scatter plotting.

## 3. Results and Discussions

### 3.1. Comparison of the Values of Vegetation Indices and SAR Variables in Each Mangrove Stage

From February 2017 to August 2020, we found no Sentinel-2 data due to cloud cover over the sample areas for approximately 18% of the days measured. As such, the number of observations for all of the variables were 9218 in total. Specifically, there were 789 observations for disturbance mangrove, 569 for recovery mangrove, and 7860 for healthy mangrove classes. The mean and standard deviation of vegetation indices and SAR backscatter variables were compared in different mangrove stages, as shown in Tables 4 and 5.

**Table 4.** The Mean and Standard Deviation of Vegetation Indices for each Mangrove Stage.

| Mangrove Stage | Vegetation Index | | | | | | | |
|---|---|---|---|---|---|---|---|---|
| | NDVI | STD ($\pm$) | NDMI | STD ($\pm$) | NDVI$_{RE}$ | STD ($\pm$) | CI$_{Green}$ | STD ($\pm$) |
| Disturbance | 0.08 | 0.11 | 0.28 | 0.20 | 0.00 | 0.04 | 1.06 | 0.22 |
| Recovery | 0.19 | 0.09 | 0.27 | 0.16 | 0.01 | 0.04 | 1.22 | 0.21 |
| Healthy | 0.53 | 0.16 | 0.32 | 0.11 | 0.08 | 0.04 | 2.05 | 0.40 |

**Table 5.** The mean and standard deviation of SAR variables for each mangrove stage.

| Mangrove Stage | VV | | | | | | VH | | | | | |
|---|---|---|---|---|---|---|---|---|---|---|---|---|
| | Ascn. | STD (±) | Descn. | STD (±) | Ascn. and Descn. | STD (±) | Ascn. | STD (±) | Descn. | STD (±) | Ascn. and Descn. | STD (±) |
| Degradation | −11.42 | 2.83 | −12.86 | 3.20 | −12.26 | 2.71 | −17.21 | 2.24 | −18.53 | 2.69 | −17.98 | 2.34 |
| Recovery | −9.84 | 2.13 | −10.48 | 2.10 | −10.23 | 1.67 | −16.10 | 1.71 | −16.65 | 1.95 | −16.43 | 1.59 |
| Healthy | −8.24 | 1.15 | −8.36 | 1.09 | −8.32 | 0.86 | −13.30 | 1.21 | −13.63 | 1.30 | −13.40 | 1.07 |

Our findings of mangrove stages using the NDVI (Table 4) were in the range of the reference data [53,55,56]. In contrast to NDMI, their values in each stage were higher than 0.1 as our references [8,9]. However, our findings of the mangrove stages using SAR backscatter variables were similar to studies by Hu. They compared mangrove forest mapping at a national scale using Sentinel-1 and Sentinel-2 time-series data on GEE, and found the mean of the NDVI was about 0.5, and the means of VH and VV backscatter on mangroves were about −14 to −15 and −8 to −10, respectively [57]. These values were similar to our healthy mangrove stages in Tables 4 and 5 and Figure 4.

In addition, a study by Cougo et al. [47] used Radarsat-2 (C-band) to model the biophysical parameters of regenerating mangroves. The incidence angle of Radarsat-2 was 23.39–25.28 degrees. They found that variable at their initial stage of mangrove, which was characterized as bare soil with a recent colonization of single seedlings of mangrove vegetation. Their mean values of VH were about −26 to −16 dB, with the mean of all of the plots being—17 dB. This compares with our disturbance mangrove stage where our finding of a VH backscatter mean was −17.98 (from multiple-incidence angles). From Cougo et al.'s advanced regeneration mangrove stage study, the trees reached 15 m in height. The VH values were about −14 to −10 dB, with the mean of all of the plots being −12 dB. This compared to our healthy mangrove stage, where our VH backscatter mean for the healthy mangrove stage from multiple-incidence angles was −13.40.

In addition, the SAR backscatter of both the VV and VH polarizations from the descending mode, which has an incidence angle of about 38.3 degrees over this study area, had a lower backscatter than the backscatter from the ascending mode (Table 5). The 2006 study on maize [61] indicated that the shallow incidence angles (>35–40 degrees) increased the path length through vegetation and maximized the vegetation scattering distribution, whereas the steep incidence angles (<30 degrees) reduced the vegetation attenuation and maximized the ground scattering contribution in return. This lower backscatter variable from a descending mode might be the result of mangrove canopy backscatter, while the backscatter variable from the incidence angle (ascending mode) was derived from the ground, mangrove roots, or stem angles. Thus, the different SAR incidence angles have no effects on dense mangrove forests. Furthermore, the standard deviation values of the backscatter from each direction are in an acceptable range of each other. The multiple-incidence angles of 32.9–38.3 degrees enhanced the correlation with the vegetation indices. Consequently, these multiple-incidence angles can be useful for future studies on mangrove stage discrimination.

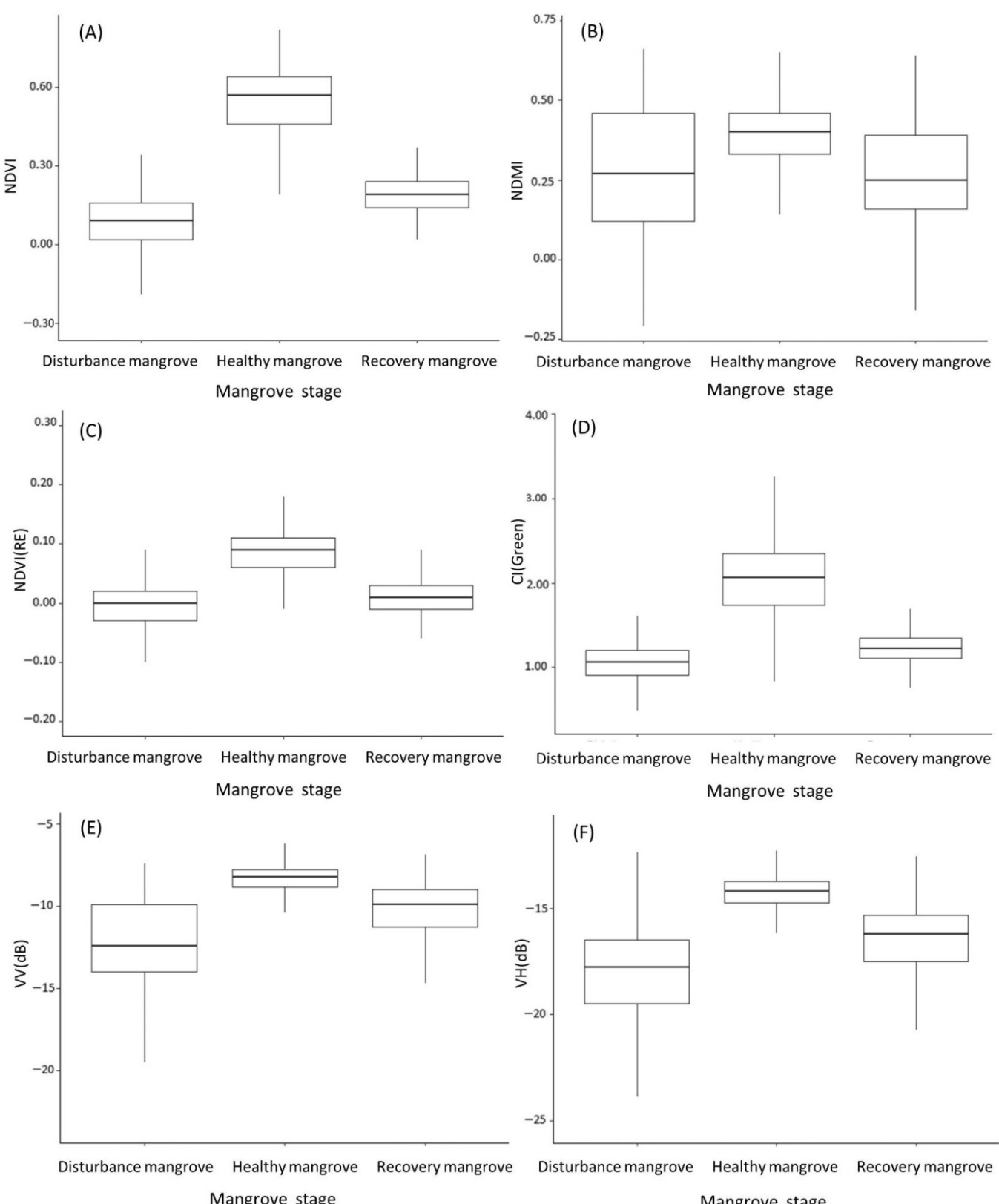

**Figure 4.** Boxplots of each mangrove stage using different vegetations indices and SAR variables. Boxplots of (**A**) NDVI, (**B**) NDMI, (**C**) NDVI$_{RE}$ (**D**) CI$_{Green}$ (**E**) VV and (**F**) VH from multiple-incidence angles for each mangrove stage.

### 3.2. Correlation Analysis: Investigating Capacity in Discriminating Mangrove Stages

3.2.1. Correlation Analysis between Vegetation Indices and SAR Variables

We found that VH had a higher correlation with the vegetation indices than VV backscatter. The SAR variables from descending fly direction (IW2) have a higher correlation with the vegetation indices compared to SAR variables from ascending fly direction (IW1) (Table 6). However, the integrated SAR fly directions could enhance the correlation with the vegetation indices. NDVI had the highest correlation with SAR variables compared to the other vegetation indices. Using Pearson's correlation, NDVI and VH from ascending and descending modes had $r^2 = 0.62$, followed by $CI_{Green}$ and VH which had $r^2 = 0.51$. The $r^2$ between NDVI and VH from the multiple-incidence angle using the polynomial regression model was 0.48, while the $r^2$ of $CI_{Green}$ and VH was 0.44.

Our results correspond to a study by Banks et al. on wetland classification using RADARSAT-2 data [62]. They found that using multiple incident angle SAR data provided a more accurate classification for all of the land covers. In addition, another study by Xu et al. using RADARSAT-2 proved that using a multi-incidence angle image produced better classification results than any single-incidence angle image for land cover classification [63].

The correlation between NDVI and VH from the multiple-incidence angles using the polynomial regression model, $r^2$ was 0.48 ($p < 0.0001$) (Figure 5). Pearson's correlation between the NDVI and VH backscatter from multiple-incidence angles revealed that $r^2$ was 0.62 ($p < 0.0001$) (Supplementary Figure S2: The Pearson's Correlation Between NDVI and VH Backscatter from Multiple-incidence Angle). A study by Veloso et al. [22] used a square of the Pearson's linear correlation coefficient to find the correlation between the temporal interpolated NDVI and SAR backscatter (VV, VH, and VH/VV) on different crops from November 2014 to December 2015. They identified the suitability of certain SAR variables for specific crops. For example, the VH/VV backscatter was poorly correlated with NDVI ($r^2$ was 0.08, $n = 15$) for a sunflower crop, while their VV correlated well with NDVI ($r^2$ was 0.77). In contrast to the sunflower, the VH/VV backscatter correlated well with NDVI ($r^2$ was 0.89, $n = 13$) for maize.

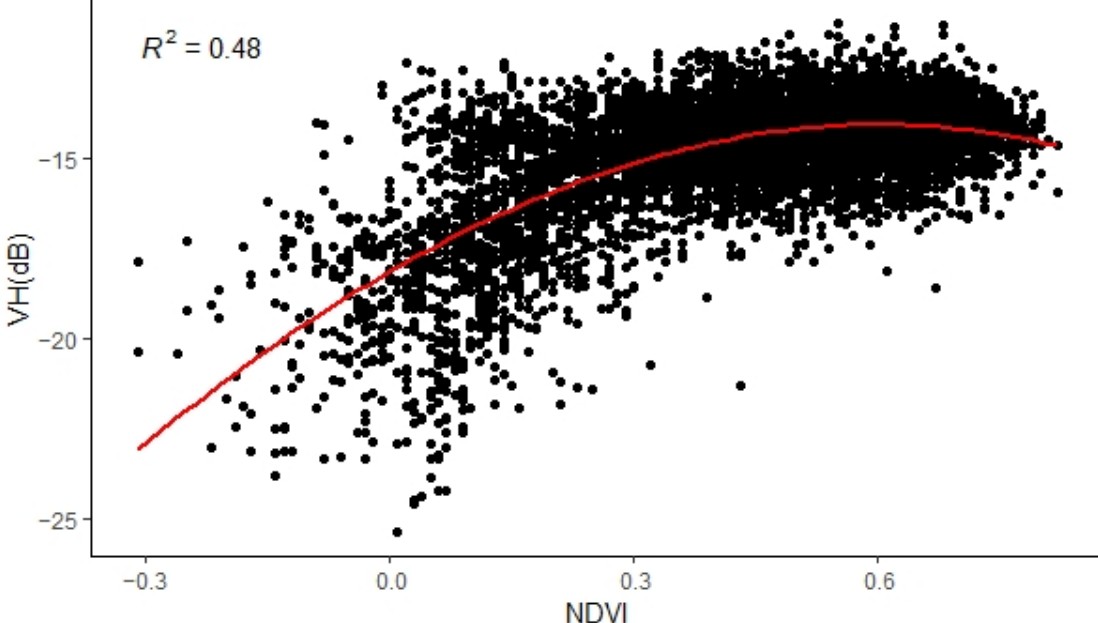

**Figure 5.** The correlation between NDVI and VH backscatter from multiple-incidence angles using the polynomial regression model.

**Table 6.** Comparison of Correlations Between SAR Variables and Vegetation Indices Using Different Models.

| Vegetation Indices | The Correlation between SAR Variables and Vegetation Indices Using Linear Regression Model | | | | | |
|---|---|---|---|---|---|---|
| | SAR Variables | | | | | |
| | VV(Ascn.) | VV(Descn.) | VV(Ascn. and Descn.) | VH(Ascn.) | VH(Descn.) | VH(Ascn. and Descn.) |
| NDVI | 0.21 | 0.33 | 0.35 | 0.26 | 0.36 | 0.38 |
| NDMI | - | - | - | −0.01 | −0.10 | −0.01 |
| NDVI$_{RE}$ | 0.10 | 0.20 | 0.20 | 0.12 | 0.22 | 0.22 |
| CI$_{Green}$ | 0.14 | 0.25 | 0.25 | 0.18 | 0.25 | 0.26 |
| | The correlation between SAR variables and vegetation indices using polynomial regression model | | | | | |
| | SAR variables | | | | | |
| | VV(Ascn.) | VV(Descn.) | VV(Ascn. and Descn.) | VH(Ascn.) | VH(Descn.) | VH(Ascn. and Desc.) |
| NDVI | 0.28 | 0.41 | 0.45 | 0.34 | 0.44 | 0.48 |
| NDMI | 0.05 | 0.04 | 0.05 | 0.05 | 0.05 | 0.06 |
| NDVI$_{RE}$ | 0.04 | 0.27 | 0.28 | 0.22 | 0.30 | 0.31 |
| CI$_{Green}$ | 0.26 | 0.40 | 0.43 | 0.32 | 0.41 | 0.44 |
| | The correlation between SAR variables and vegetation indices using Pearson's model | | | | | |
| | SAR variables | | | | | |
| | VV(Ascn.) | VV(Descn.) | VV(Ascn. and Descn.) | VH(Ascn.) | VH(Descn.) | VH(Ascn. and Descn.) |
| NDVI | 0.45 | 0.57 | 0.59 | 0.51 | 0.60 | 0.62 |
| NDMI | 0.01 | 0.00 | 0.01 | 0.08 | 0.10 | 0.10 |
| NDVI$_{RE}$ | 0.31 | 0.45 | 0.44 | 0.37 | 0.47 | 0.47 |
| CI$_{Green}$ | 0.38 | 0.48 | 0.50 | 0.42 | 0.50 | 0.51 |

In addition, we investigated the correlation between NDVI and VH from the multiple-incidence angles on the plot ID2 (data available in Supplementary Figure S3: Data Available of Plot ID2 From Sentinel-1 SAR And Optical Data from Sentinel-2 During February 2017 to August 2020), which represented the mangrove's change from healthy to disturbance stage (Figure 6). We found that the Pearson's correlation $r^2$ was 0.93 ($p < 0.0001$, $n = 34$), while the $r^2$ was 0.91 using a polynomial regression model. Compared to overall observation ($n = 9258$), a decrease in the correlation of the large observations might result from the variation of both optical and SAR backscatter.

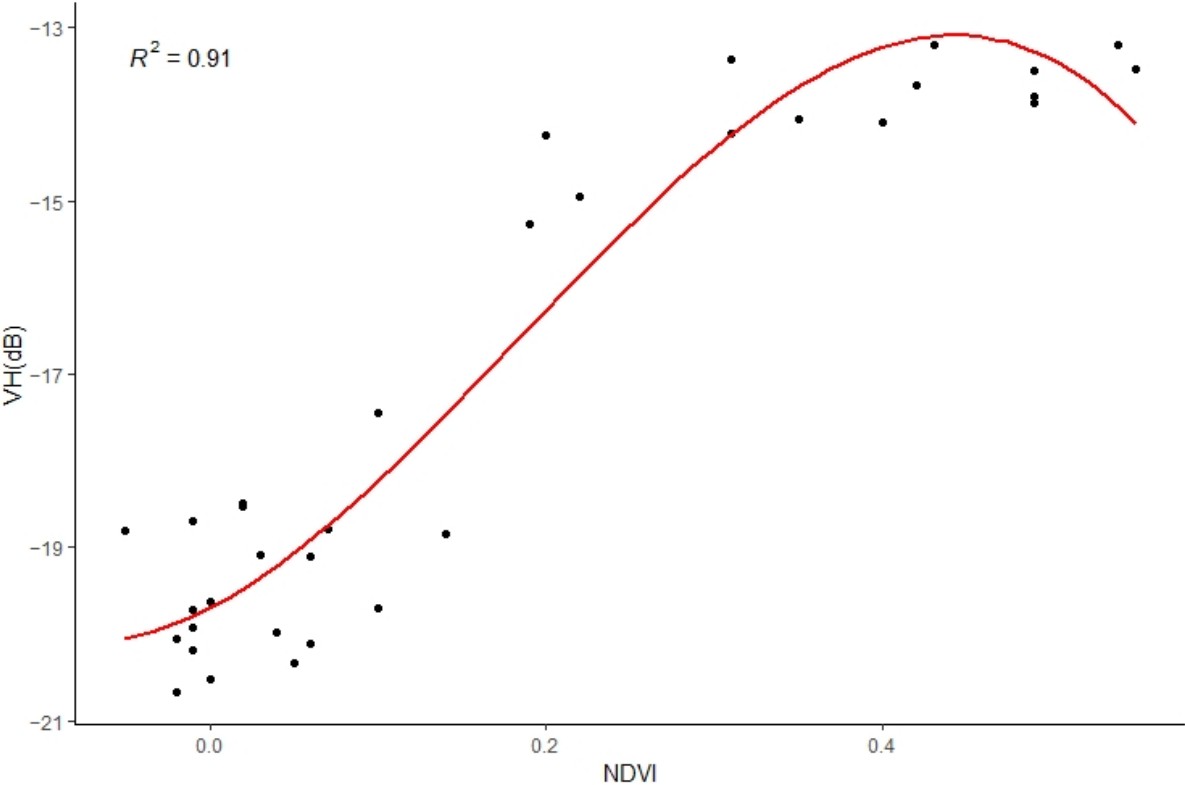

**Figure 6.** Correlation between NDVI and VH backscatter from multiple-incidence angles using the polynomial regression model on plot ID2.

### 3.2.2. Investigating Capacity of Vegetation Indices in Discriminating Mangrove Changes

We found that NDVI had the highest correlation, while NDMI had lower correlations with the SAR backscatter variables than other vegetation indices. Thus, all of the vegetation indices were then investigated using their time series. Their values were revealed due to the mangrove stage changes witnessed therein. Here, we selected two plots to represent the mangrove stage changes in the time series. The first plot is plot ID2, representing the healthy mangroves that were changed to a disturbance mangrove stage (Figure 7). The second plot was plot ID376. It represented a disturbed mangrove that had been changed to a healthy mangrove stage (Figure 8). These illustrated that NDMI had significantly varied with no (or less) vegetation cover in comparison to other vegetation indices. In some periods, the NDMI values were still higher than the disturbed mangrove forest. However, the NDMI values on the dense mangrove cover had no significant fluctuation. On the other hand, NDVI, NDVI_{RE} and CI_{Green} had a similar pattern of mangrove change. They had the same reduced values, while NDMI had increasing values, as shown in purple dash circles (which might have resulted from a high sea level or soil moisture). Consequently, NDMI was deemed unsuitable for discriminating the mangrove stages, especially during times of poor vegetation cover. This finding was contrary to other studies which found that NDMI could detect disturbance vegetation better than NDVI [5,64].

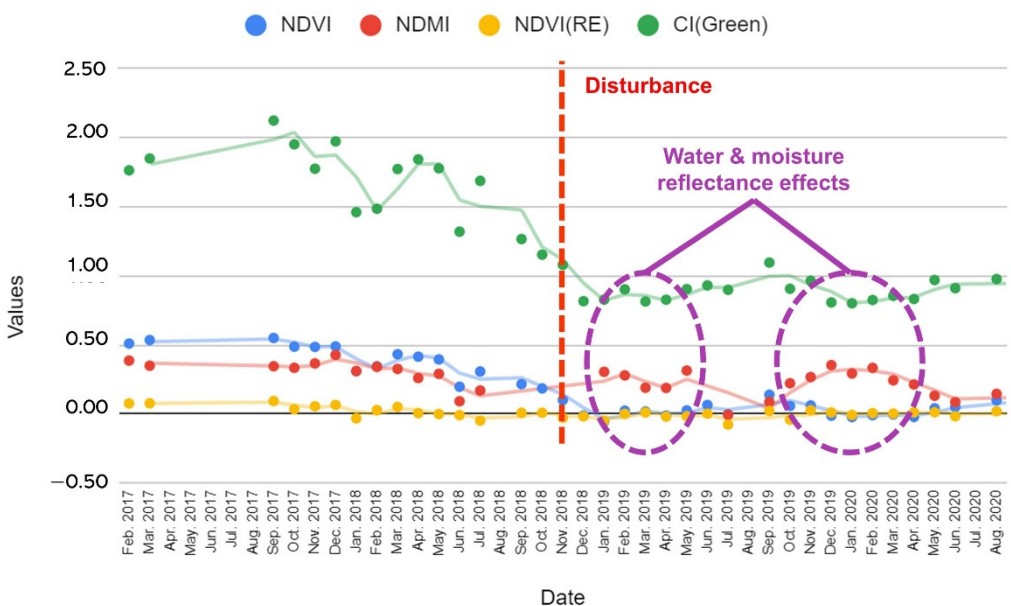

**Figure 7.** Comparison of Mangrove Vegetation Indices—Stage Change from Healthy to Disturbance using Sample Plot ID2. Disturbance to healthy mangrove stage using the sample plot ID376.

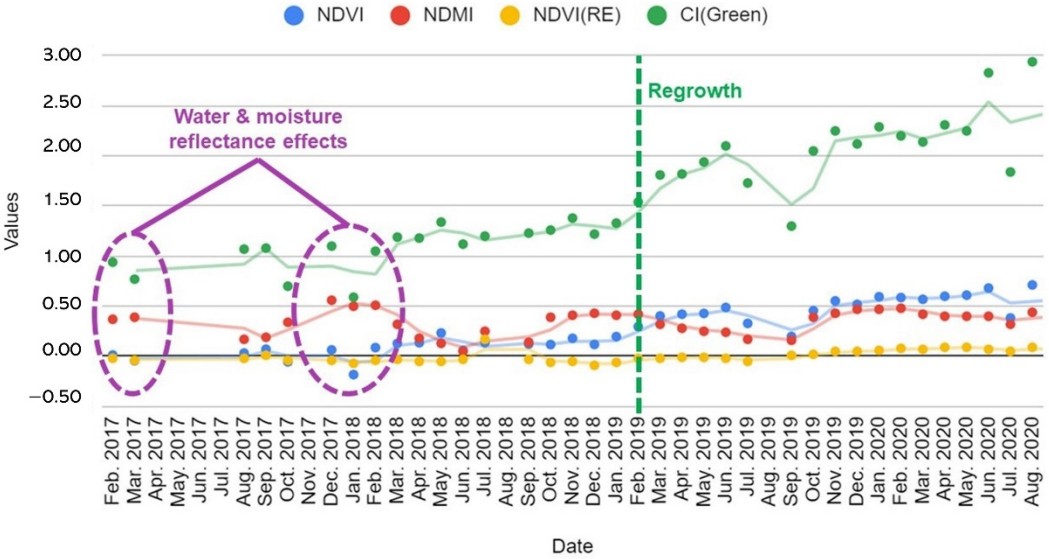

**Figure 8.** Comparison of Mangrove Vegetation Indices—Stage Change from Disturbance to Healthy using Sample Plot ID376.

### 3.3. Rule-Based Analysis of the Highest Correlation Variable and the Capacity for Discriminating Mangrove Stages

3.3.1. Rule-Based Analysis on NDVI Time-Series of Different Mangrove Stages

NDVI was selected to analyze value changes using the rule-based methodology on the time-series of different mangrove stages because it had the highest correlation with VH. This analysis used Sentinel-2 data dating back to January 2016. This was the available data that covered our study area and was most suitable for understanding mangrove stage changes over a longer period.

The sample plot ID87 represented mangrove stage changes, including disturbance, recovery, and healthy mangroves (Figure 9). The observations were labeled following the rule-based methodology. The representative symbols of disturbance, recovery, and healthy mangroves were defined by rule-based red, green, and blue dots. The red, green, and blue

lines were representative of disturbance, recovery and healthy mangrove stages defined by the time aspect. We can see that some of the observations did not match the rule-based system when they were coupled with time. Indeed, five observations did not match with the rule-based methodology. Considering the time aspect, one errant observation was in the recovery mangrove class. It had an NDVI value higher than 0.30, and would be considered a healthy variable with respect to the rule-based system. Three other time aspect observations in the healthy class did not match the rule as their values were less than 0.30. In addition, one observation in the healthy mangrove class (based on time/blue line) had an NDVI value less than 0.10; this would put it into the disturbance class under the rule-based methodology.

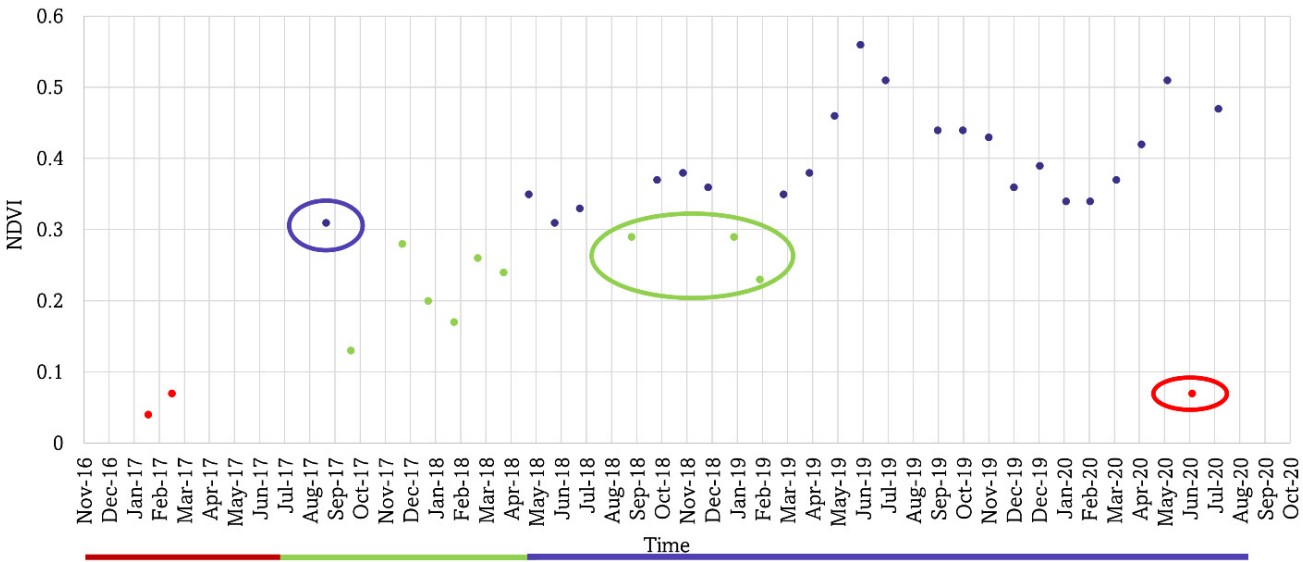

**Figure 9.** NDVI Time-Series Representative Plot ID87—Mangrove Stage Changes Using Rule-based Methodology Degradation to Recovery; The Circled Showed Some NDVI of Each Mangrove Stage Did Not Follow NDVI Rule-Based on Time-series Data.

The sample plot ID148 represented a healthy mangrove that was changed to a disturbance mangrove. Regarding time (Figure 10), this plot was considered to be a healthy mangrove until December 2019 (blue line). Using the rule-based methodology, this plot shows that some of the values were labeled as a disturbance as they were less than 0.30. After December 2019, with regards to time (red line), the mangrove stage became disturbance. However, some observations were labeled as healthy as their values were higher than 0.30.

The mismatch between the rule-based and time methodologies in identifying the mangrove stages was approximately 20% of all observations. The total number of observations was 9258. When using a rule-based labeled mangrove disturbance stage on time-series, there were 6 mismatches from 789 observations or 0.76%. These observation values in the time series were labeled as a disturbance, but their values were higher than 0.3 in respect to rule-based values.

For the recovery mangrove stage, there were 43 mismatches from 569 observations or 7.56%. These were labeled as disturbances using the rule-based methodology but were not matched with time-series. As in time-series, the mangrove stage was recovery, but their values were less than 0.10. In addition, the NDVI values greater than 0.29 numbered 34 from 569 total observations using the rule-based methodology; or a 5.98% mismatch to the healthy mangrove class based on the time-series. Therefore, the mismatched values totaled 13.54% for the recovery class using the rule-based methodology on time-series. For the healthy mangrove stage, there were 434 mismatches from 7860 observations (which were less than 0.30) or a 5.52% mismatch with healthy class values based on the time methodology.

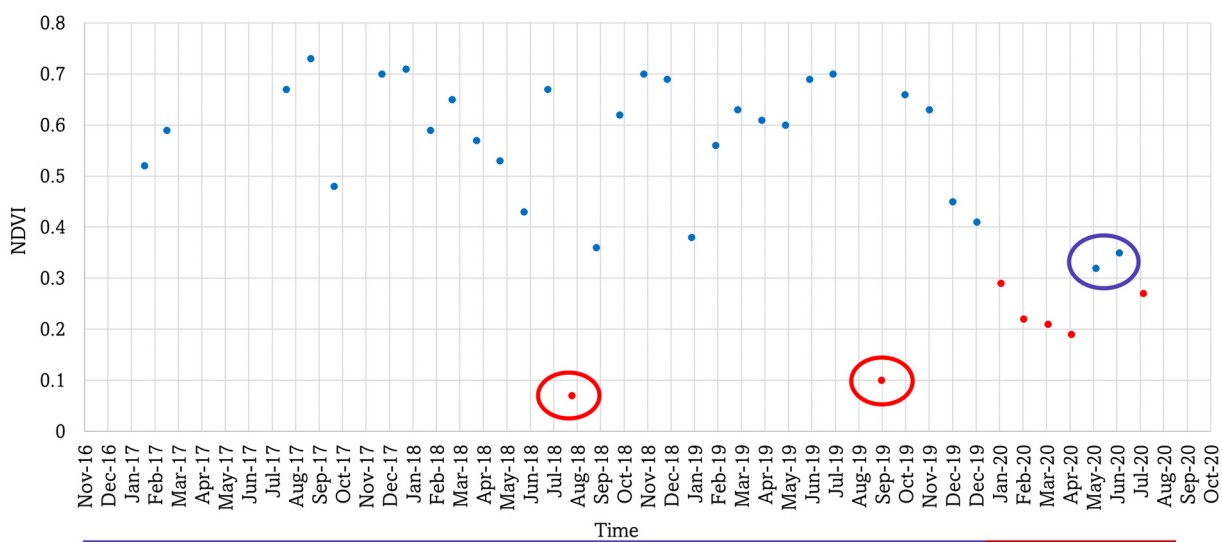

**Figure 10.** NDVI Time-Series Representative Plot ID148—Mangrove Stage Changes Using Rule-based Methodology Healthy to Degradation; The Circled Showed Some NDVI of Each Mangrove Stage Did Not Follow NDVI Rule-Based on Time-series Data.

As was determined by Li et al. [21], the NDVI time-series data had some variations resulting from atmospheric conditions and other effects. Some of the values suddenly dropped whilst some peaked when the nearby month values were stable on the trend line. Although we masked the clouds out, these uncertainties led to undesirable noise. In addition, the mangrove stage changes take time to develop due to changes in their biomass or canopy cover.

### 3.3.2. Analysis of the Capacity of NDVI and VH from Multiple-Incidence Angles in Discriminating Mangrove Stage Changes

We analyzed the NDVI capacity for discriminating mangrove stage changes in correlation with VH variables from multiple-incidence angles. The plots ID145 and ID307 illustrated the mangrove stage changes from a healthy stage to a disturbance stage using NDVI (A) and VH from multiple-incidence angles (B). Plots ID12 and ID398 illustrated a mangrove stage change from disturbance to recovery; then, on to a healthy stage using NDVI (C) and VH from multiple-incidence angles (D) (Figure 11). Furthermore, we found that NDVI and VH in plot ID145 had higher values during the healthy mangrove stage than seen in plot ID307 (in the blue dash square); however, using VH gave more variations than using NDVI. In the blue dash square, the ID12 and ID398 plots' VH values were also similar during the healthy stage (D), but the values of these two plots showed distinct differences using NDVI (C). Thus, it is concluded that NDVI displayed better discriminating capacity for mangroves in the healthy stage in comparison with VH.

During the disturbance mangrove stage, we found that VH from plot ID145 did not reduce clearly compared to NDVI. It showed higher VH fluctuations during this period (B). We then investigated the mangrove stages using high spatial resolution on Google Earth Pro. We found that after the mangroves were disturbed, some of the mangrove stems and branches were still on the soil or bare land (Supplementary Figure S4: The Satellite Images from Google Earth Pro Before and After Mangrove Forest Was Disbanded on Plot ID145). These could result in high variations on VH readings in the red dash square (B). The SAR C band could still detect tree trunks, stems, or branches even when the canopy was reduced [65]. Contrary to plot ID307, VH reduced and corresponded with NDVI, where stems (or trunks) were already removed and the land converted to ponds or otherwise covered with water (Supplementary Figure S5: The Satellite Images from Google Earth Pro Before and After Mangrove Forest Was Disbanded On Plot ID307).

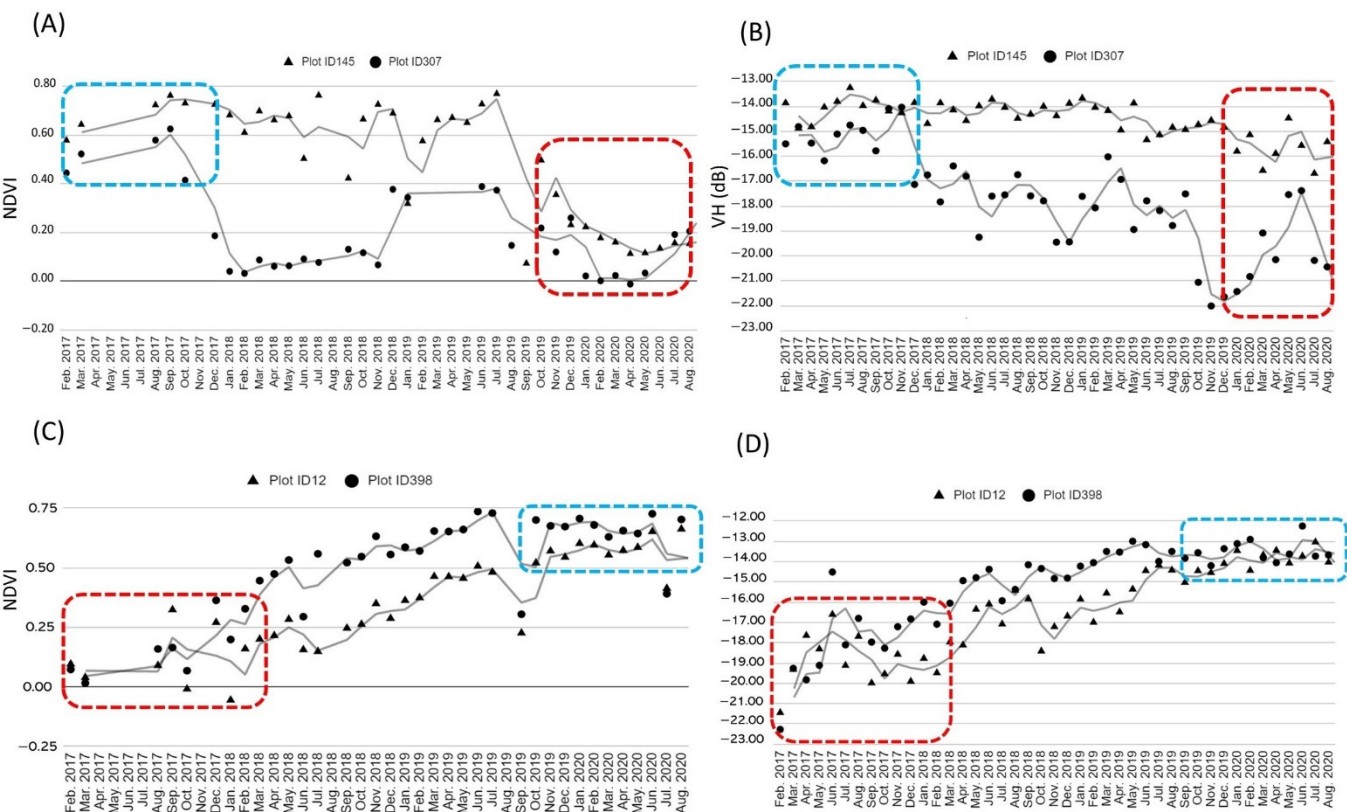

**Figure 11.** Example plots of mangrove stage changes. (**A**) Two sample plots of healthy mangrove changing to disturbance mangrove stages using NDVI, (**B**) Two sample plots of healthy mangrove changing to disturbance mangrove stages using VH from multiple-incidence angles, (**C**) Two sample plots of disturbance mangrove changing to recovery and on to healthy mangrove stages using NDVI, (**D**) Two sample plots of disturbance mangrove changing to recovery and on to healthy mangrove stages using VH from multiple-incidence angles, The red rectangles showed the lower period of mangrove forest covering, while the blues rectangle showed the high period of mangrove covering.

Meanwhile, we found that the trends from both of the variables revealed similar increasing values during the recovery mangrove stages (C and D). Both NDVI and SAR variables could be employed to discriminate the recovery mangroves. However, before the mangroves recovered, a reduced mangrove canopy illustrated high fluctuation in NDVI (C) and VH (D) in the red dash squares. This resulted from moisture, tidal flow, or other waterborne effects [66].

Next, the correlation between NDVI and VH from multiple-incidence angles was analyzed using clusters of different mangrove stages (Figure 12). We found that both the disturbance and recovery mangrove stages overlapped with the healthy mangrove stage in density scatter plots of NDVI (A) and VH (B). These were caused by using the rule-based methodology on time-series data. We can deduce that both the NDVI rule-based system and VH backscatter on time-series data had some variations resulting from atmospheric variability and other effects [21,66]. Therefore, using only the rule-based methodology (or the algorithms which are not concerned with their time series counterpart) would not be adequate to classify the mangroves from each other. They had overlap values that could not be classified accurately using a single scene of satellite data.

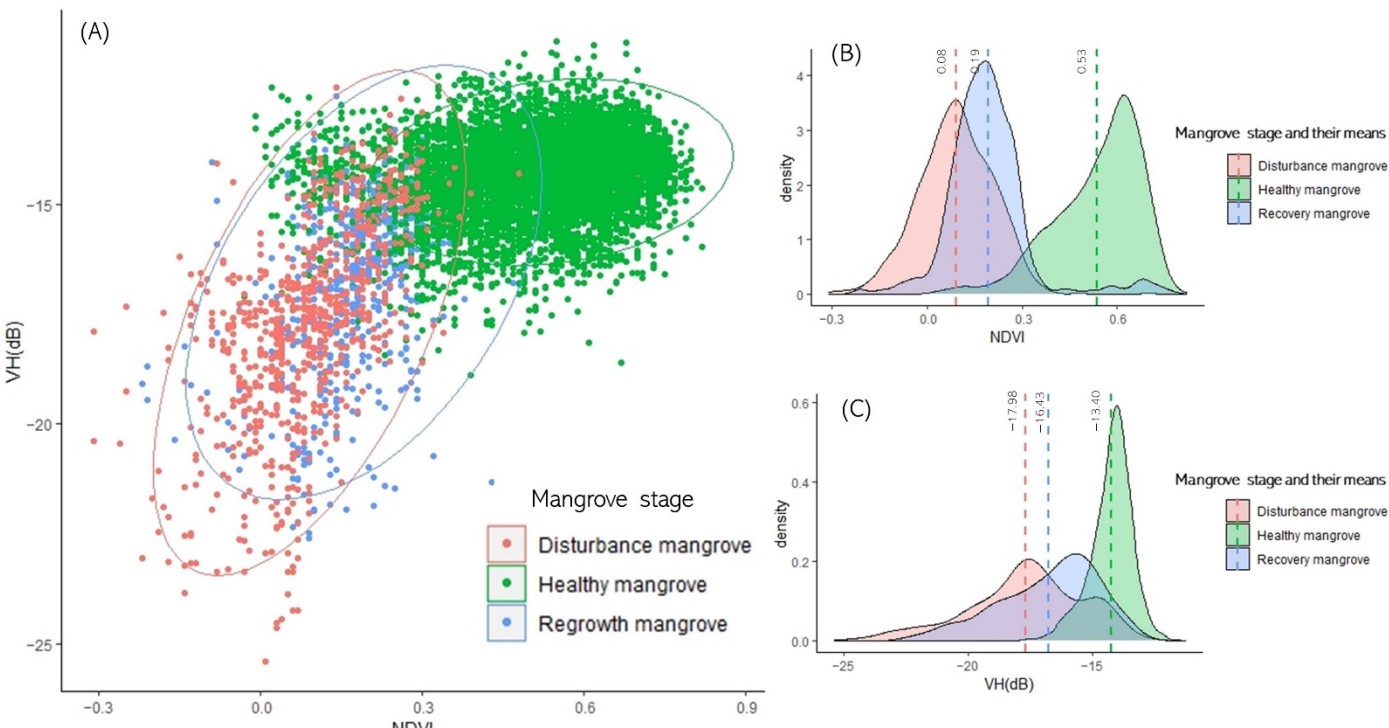

**Figure 12.** Clusters and Density Scatter Plotting of Mangroves in Different Stages Using the Correlation of NDVI and VH from Multiple-Incidence Angles. (**A**) The correlation between NDVI and VH backscatter from multiple-incidence angles illustrating the cluster of each mangrove stage. (**B**) The density scatter plot of NDVI for each mangrove stage. (**C**) The density scatter plot of VH for each mangrove stage.

NDVI and VH could both be used to discriminate the mangrove stages, with VH reaching saturation faster when the canopy is well-developed. In situations where the mangrove trunks, stems, and branches were still on the ground, the VH values were not recommended. Rather, the use of NDVI was suggested instead. Both NDVI and VH could be used to discriminate in the recovery mangrove stage as they accurately revealed the developing changes (C and D). However, it should be noted that NDVI (from the optical sensor) and VH backscatter (from SAR C-band) will exhibit variations when discriminating mangrove stages affected by environmental factors. This is due to their design capacity or sensitivity to external factors. In these circumstances, applying moving average calculations on the NDVI and VH data should enhance their accuracy in mangrove discrimination.

Therefore, multi-temporal NDVI, NDVI$_{RE}$ and CI$_{Green}$ show promising capacity in discriminating the mangrove stage including disturbance, recovery, and healthy mangrove stages. Unfortunately, about 20% of the optical data from Sentinel-2 are not available due to cloud cover. This limits the continuous monitoring of the mangrove forest stage changes [18]. If the status of the mangrove is recently known, it could prevent the loss of the mangroves and their ecosystem. Here, we found that SAR backscatter can be used to fulfill the uncertainty of unavailable optical data as there are correlations with the vegetation indices. VH from multiple-incidence angles is recommended for discriminating the mangrove stages changes to reduce the bias for monitoring time-series data, as the different incidence angle of SAR data detects different parts of mangrove trees.

The spatial map of the mangrove forest stage changes during February 2017 to July 2020, which are discriminated using the NDVI rule-based method correlated with VH backscatter from multiple-incidence angles, is shown in Figure 13. VH backscatter is used to fill the gap of cloud cover over the mangrove forest. The image on July 2020 is selected to illustrate the cloud cover over the mangrove forest and the use of combine NDVI and VH backscatter.

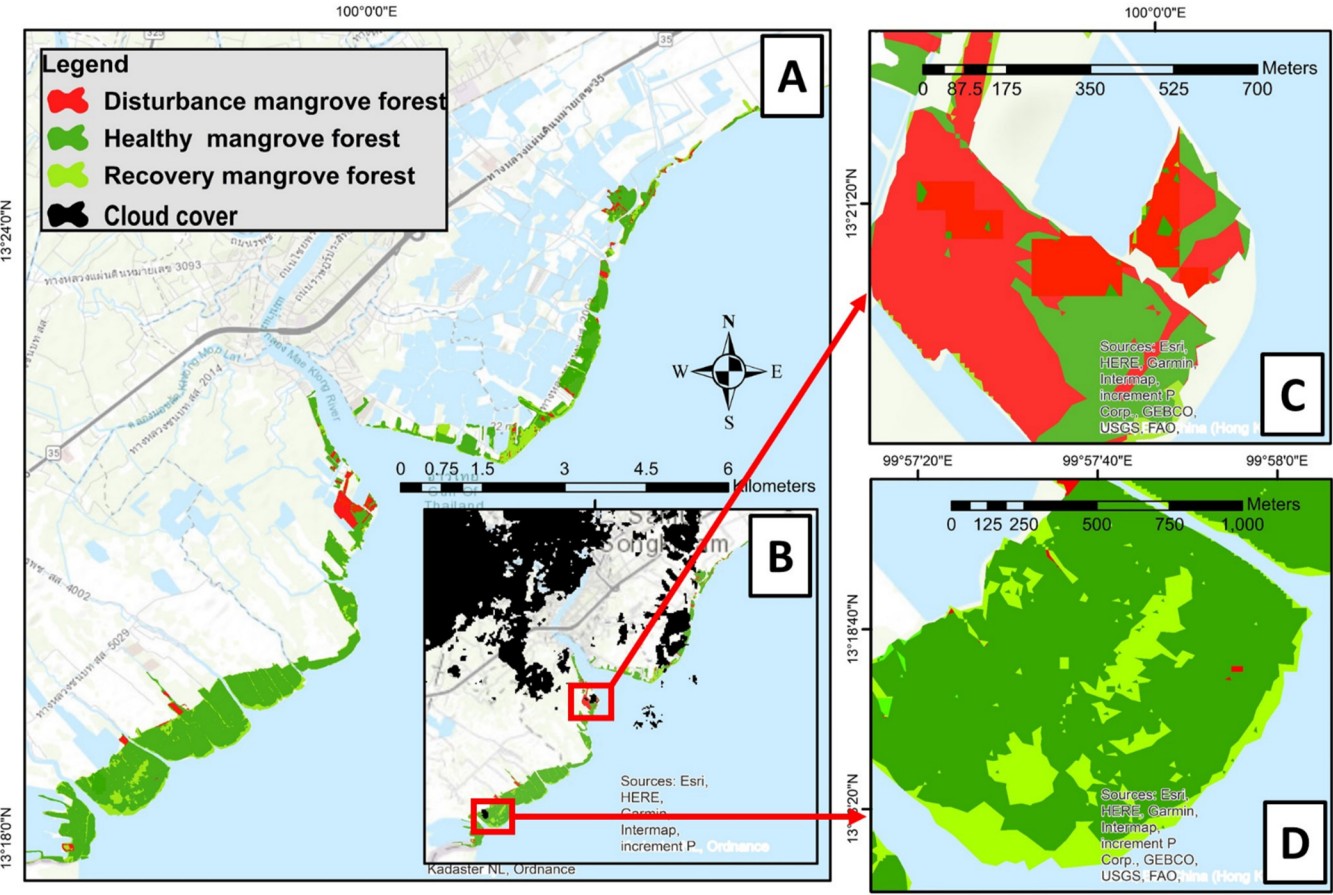

**Figure 13.** Mangrove Stage Change During February 2017 to July 2020 Using NDVI Rule-based Correlated with VH Backscatter from Multiple-incidence Angles (**A**) Mangrove Forest Stages in Samut Songkhram Province (**B**) Cloud Cover Over Mangrove Forest During July 2020 (**C**) Sample Area of Disturbance Mangrove Stages using NDVI Rule-based Correlated with VH Backscatter (in Red) (**D**) Sample Area of Recovery Mangrove Stages using NDVI Rule-based Correlated with VH Backscatter (in Light Green).

Our findings of the SAR backscatter values from the effects of different multiple-incidence angles correspond with previous studies [62,63]. In addition, this study found that the SAR backscatter can discriminate the recovery mangrove stage well, especially in the early stages of recovery because it can detect even the mangroves' stems; thus, this benefit of SAR backscatter gives an opportunity to discriminate the mangrove stage changes continuously. The recent information on mangrove stages can introduce early rehabilitation which can complete ecological loss [67,68].

However, SAR backscatter has limitations in the discrimination disturbance mangrove stages during the early disturbance mangrove stages, such as stress in the mangroves or crown dieback, because SAR can still detect the branches or stems of mangrove trees. Future studies might use other new Radar Vegetation indices. For example, a study by Mandal et al. (2020) [69] found that DpRVI (compared with the cross and co-pol ratio ($\sigma$VH0/$\sigma$VV0) and dual-pol Radar Vegetation Index (RVI = 4$\sigma$VH0/($\sigma$VV0 + $\sigma$VH0)) has a high correlation with biophysical parameters (including Plant Area Index (PAI), Vegetation Water Content (VWC), and dry biomass (DB)) of a canola crop. The new vegetation index from Sentinel-1 might enhance capacity in discriminating the mangrove stage in early disturbance mangrove stages.

## 4. Conclusions

Using data from Sentinel-1 SAR backscatter and Sentinel-2, we found that NDVI and VH could discriminate the mangrove stage changes. The disturbance, recovery, and healthy mangrove stages determined by NDVI values were approximately 0.08 ($\pm$0.11), 0.19 ($\pm$0.09), 0.53 ($\pm$0.16), respectively. VH backscatter values for disturbance, recovery, and healthy stages from multiple-incidence angles were approximately $-17.98$ ($\pm$2.34), $-16.43$ ($\pm$1.59), $-13.40$ ($\pm$1.07), respectively. The correlation between NDVI and VH from the multiple-incidence angles using Pearson's correlation was $r^2 = 0.62$ ($n = 9258$).

It is possible to conclude that using NDVI is best for detecting disturbance mangroves in their early stages, VH is best for detecting the recovery stage of mangroves because, if there is a mixing reflectance from the soil, water, and moisture that can skew the NDVI values, and both NDVI and VH have the capacity to detect healthy mangroves. Remarkably, the high sensitivity of SAR could detect such small details as the stems of newly generated mangroves. We found that the combined optical data from Sentinel-2 and SAR data from Sentinel-1 can be analyzed and used to discriminate mangrove forest change stages.

Moreover, our findings on the correlation between multitemporal NDVI and VH should benefit timely and efficient mangrove management and contribute to sustaining mangrove services. The mangrove forest areas in their individual conditional stages can be addressed using both of the variables. This can supplement the data of optical sensors when such data are unavailable. This is particularly useful in early mangrove rehabilitation from general stress or at the disturbance stage. It could prevent the loss of mangrove biomass and quicken mangrove restoration. Prompt and accurate information on mangrove stages, and the changes incurring within, will aid appropriate remedial policies and ultimately lead to knowledge-based decision-making on sustainable conservation and the competent management of this precious natural resource.

Time-series data from both optical and SAR data can be analyzed and interpolated to enhance the mangrove stage discrimination's accuracy. Even better, machine learning or deep learning can be employed to expedite the analysis.

**Supplementary Materials:** The following supporting information can be downloaded at: https://www.mdpi.com/article/10.3390/f13091433/s1, Figure S1: Sentinel-1 C-band Fly Directions Over the Study Area, Figure S2: The Pearson's Correlation Between NDVI and VH Backscatter from Multiple-incidence Angle; Figure S3: Data Available Of Plot ID2 from Sentinel-1 SAR And Optical Data from Sentinel-2 During February 2017 To August 2020; Figure S4: The Satellite Images from Google Earth Pro Before and After Mangrove Forest Was Disbanded on Plot ID145; Figure S5: The Satellite Images From Google Earth Pro Before and After Mangrove Forest Was Disbanded On Plot ID307; Table S1: The Specification of Sentinel-2 Data.

**Author Contributions:** Conceptualization, K.U.; methodology, S.N. and N.S.; validation, K.U.; formal analysis, K.U.; writing—original draft preparation, K.U.; writing—review and editing, S.N., S.G.P.V. and N.S.; visualization, S.N., S.G.P.V. and N.S.; supervision, S.N., S.G.P.V. and N.S. All authors have read and agreed to the published version of the manuscript.

**Funding:** This research received no external funding.

**Institutional Review Board Statement:** Not applicable.

**Informed Consent Statement:** Not applicable.

**Data Availability Statement:** Not applicable.

**Acknowledgments:** The authors would like to acknowledge Geoinfo Co., Ltd., Thailand for supporting coding for extracting time-series data of vegetation indices from Google Earth Engine.

**Conflicts of Interest:** The authors declare no conflict of interest.

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
