# Peer review of "Discrimination of Mangrove Stages Using Multitemporal Sentinel-1 C-Band Backscatter and Sentinel-2 Data—A Case Study in Samut Songkhram Province, Thailand"

_forests, doi:10.3390/f13091433_

Round 1

Reviewer 1 Report

Authors have chosen very special topics which will be interest to readers. There are several flaws in the manuscript. Although the problem being addressed is definitely of interest to  readers and the methodology is little bit okay , your manuscript does not meet the required quality standards for consideration as per present quality and writing standard. 

1. Abstract does not have sufficient parameters, like indices utilized in studies- only one is shown while four is being used for research.  at many points, sentences are not reflecting the correct meaning , what you wanted to convey to your readers. 

2. The Introduction does not cover the pertinent literature and fails to highlight the novelty of your study. 

3. Introduction does not reflect the optical failure which you mentioned in abstract. there are several studies which are conducted earlier to demonstrate the robustness of optical data- either multi-or hyperspectral-  

2019 Spatial distribution of mangrove forest species and biomass assessment using field inventory and earth observation hyperspectral data. Biodiversity and Conservation28(8), pp.2143-2162.

 2021. Spectral Complexity of Hyperspectral Images: A New Approach for Mangrove Classification. Remote Sensing13(13), p.2604.

2020 Use of hyperion for mangrove forest carbon stock assessment in Bhitarkanika forest reserve: A contribution towards blue carbon initiative. Remote Sensing12(4), p.597.

 2018. Hyperspectral estimation of the chlorophyll content in short-term and long-term restorations of mangrove in Quanzhou Bay Estuary, China. Sustainability10(4), p.1127.

4. Your method description is satisfactory, but far from being clear to a broader audience. 

5. You have implemented 4 indices which are very common, why not use more indices from sentinel 1 data, vegetation indices from microwave data will be useful to present your results.  As without these it will be useless to incorporate microwave data in your studies. 

6. Provide the spatial maps of mangrove forest , to illustrate the disturbed regions, recovered regions. final map is required to show how it has been identified and mapped. 

7.  The Discussion is non-existent and fails to draw parallels with relevant studies from the peer-reviewed literature or put the presented results into the right context. I encourage authors to discuss results with previous studies or in related contexts. 

8.  Scientific writing style is altogether missing throughout the manuscript. I would suggest authors to see the entire manuscript thoroughly and rephrase / modify sentences to convey meaningful information to readers. 

Author Response

Could you please see the attachments of response and MS with track changes?

Thank you very much

Reviewer 2 Report

The manuscript "Discrimination of Mangrove Stages Using Multitemporal Sentinel-1 C-Band Backscatter and Sentinel-2 Data – A Case Study in Samut Songkhram province, Thailand" investigates the ability of Sentinel-2 derived vegetation indices and Sentinel-1 derived radar backscatter to discriminate between stages of mangrove growth in Thailand. The authors focus their discussion on the quantitative aspect of their findings and compare their results to what others found using similar techniques in mangrove ecosystems in other areas. While it is good to know the agreements and disagreements, I would suggest the authors to focus more on the underlying ecological theory to explain their results.

Major Comments:

1. Overall writings needs to be more compact to improve readability.

2. A greater integration of ecological theory of mangrove growth and remotely sensed indices/backscatter is suggested to explain the results.

3. Line 134. Why was Sentinel-2 TOA used instead of Surface Reflectance data? TOA is not atmospherically corrected which may introduce errors in a time-series. Did the authors perform any atmospheric correction on the TOA images?

4. Line 141. Was the cloud masking step using Sentinel-2 bands 10 and 11 effective? Any additional steps taken to completely mask clouds? 

5. Line 149-150. "Next, we used zonal statistics to extract the averaged 10 maximum random values of vegetation indices. "

Explain this step and its objectives in more details. The purpose of this step is unclear.

6. Line 154. Why was Sentinel-1 data from GEE not used? If I understand correctly, the processing of Sentinel-2 data was done in GEE.

Minor Comments:

1. Line 36. Reference and the year of introduction of the National Economic and Social Development Plans (NESDP) is missing.

2. Section 2.2.1 Defining Polygon Samples of Different Mangrove Stages on Google Earth Pro- Providing photos or quantitative satellite based metrics of how the different mangrove stages were separated is recommended.

3. Line 135. "January 2016 to August" - Incomplete dates.

Author Response

Could you please see the attachment of the response and MS with track changes?

Thank you very much.

Round 2

Reviewer 1 Report

The authors did a lot of good work in the current version and incorporated all suggestions. The current revised MS version has been significantly improved in scientific writing and as per guidelines. In my opinion, MS is suitable for publication in Forests.